# Effects of Consuming Ounce-Equivalent Portions of Animal- vs. Plant-Based Protein Foods, as Defined by the Dietary Guidelines for Americans on Essential Amino Acids Bioavailability in Young and Older Adults: Two Cross-Over Randomized Controlled Trials

**DOI:** 10.3390/nu15132870

**Published:** 2023-06-25

**Authors:** Gavin Connolly, Joshua L. Hudson, Robert E. Bergia, Eric M. Davis, Austin S. Hartman, Wenbin Zhu, Chad C. Carroll, Wayne W. Campbell

**Affiliations:** 1Department of Nutrition Science, Purdue University, West Lafayette, IN 47907, USA; connolg@purdue.edu (G.C.);; 2Department of Statistics, Purdue University, West Lafayette, IN 47907, USA; 3Department of Health and Kinesiology, Purdue University, West Lafayette, IN 47907, USA

**Keywords:** dietary protein, protein recommendations, amino acid composition, protein quality, metabolism, protein synthesis, anabolic response, muscle, aging, ageing

## Abstract

Background: The Dietary Guidelines for Americans (DGA) recommends consuming a variety of “Protein Foods” based on “ounce-equivalent” (oz-eq) portions. No study has assessed the same oz-eq portions of animal- vs. plant-based protein foods on essential amino acid (EAA) bioavailability for protein anabolism in young and older adults. Objectives: We assessed the effects of consuming two oz-eq portions of pork, eggs, black beans, and almonds on postprandial EAA bioavailability in young and older adults. Methods: We conducted two investigator-blinded, randomized crossover trials in young (*n* = 30; mean age ± SD: 26.0 ± 4.9 y) and older adults (*n* = 25; mean age ± SD: 64.2 ± 6.6 y). Participants completed four testing sessions where they consumed a standardized meal with two oz-eq of either unprocessed lean pork, whole eggs, black beans, or sliced almonds. Blood samples were taken at baseline and 30, 60, 120, 180, 240, and 300 min postprandially. Plasma EAA bioavailability was based on postprandial integrated positive areas under the curve. Results: Participant age did not affect EAA bioavailability among the four protein foods tested. Two oz-eq portions of pork (7.36 g EAA) and eggs (5.38 g EAA) resulted in greater EAA bioavailability than black beans (3.02 g EAA) and almonds (1.85 g EAA) in young and older adults, separately or combined (*p* < 0.0001 for all). Pork resulted in greater EAA bioavailability than eggs in young adults (*p* < 0.0001), older adults (*p* = 0.0007), and combined (*p* < 0.0001). There were no differences in EAA bioavailability between black beans and almonds. Conclusions: The same “oz-eq” portions of animal- and plant-based protein foods do not provide equivalent EAA content and postprandial bioavailability for protein anabolism in young and older adults.

## 1. Introduction

The 2020–2025 Dietary Guidelines for Americans (DGA) recommend that “Protein Foods” should be consumed as part of a healthy dietary pattern [1]. The protein foods group includes animal-based—red meats, poultry, fish, and eggs—and plant-based—soy products, beans, peas, and lentils, and nuts and seeds—protein-rich foods. The DGA uses ounce-equivalent (oz-eq) as the unit of measure to “identify the amount of (protein group) foods… with similar nutritional content” [1]. For example, one oz-eq equals one ounce of meat, one whole egg, 0.25 cups of beans, or 0.5 ounces of nuts. The basis for stating that these protein foods are “equivalent” and have “similar nutritional content” is unclear. Protein foods differ in metabolizable energy (up to 3-fold) and macronutrient contents, including protein quantity and quality. For example, regarding protein quantity, one oz-eq of pork loin contains ~7 g of total protein, whereas one oz-eq of almonds contains ~3 g of total protein. Protein quality can be defined as the ability of a dietary protein source to provide adequate amounts and proportions of essential amino acids (EAA)—which can only be obtained via dietary consumption—that are digestible and subsequently become bioavailable for use in the body for metabolic processes, such as stimulating protein synthesis and maintaining or growing body tissues [2,3]. Animal-based protein foods contain all the EAA required by humans, whereas plant-based protein foods, other than soy, are lacking in one or more EAA [2,3,4]. The EAA content of a protein-containing food or a meal determines the anabolic response to feeding [5,6,7], which chronically may impact tissue and whole-body metabolism, morphology, and health [8,9,10].

Research by Park et al. [6] with young adults found that consuming two oz-eq of animal-based protein foods resulted in a greater whole-body net protein balance than plant-based protein foods. Among all food sources (each consumed without other foods), a greater net protein balance (anabolic) response was correlated with higher food EAA content and greater whole-body protein synthesis [6]. Limited information exists regarding the effects of consuming equivalent amounts of protein foods from different sources as part of a mixed meal on postprandial EAA bioavailability. The 2020–2030 Strategic Plan for NIH Nutrition Research includes a goal to “define the role of nutrition across the lifespan” with an objective to “assess the role of nutrition in older adults to promote healthy aging” [11]. However, there is a paucity of primary research that directly compares EAA bioavailability between young and older adults consuming the same oz-eq portions of varied protein foods.

Therefore, we conducted two sequentially randomized, investigator-blinded, crossover acute feeding trials with the same study design: the first in a cohort of young adults and the second in a cohort of older adults. The primary objective of this project was to assess the effect of consuming two oz-eq portions of animal-based (unprocessed lean pork or whole eggs) vs. plant-based (black beans or sliced almonds) protein foods as part of a mixed whole foods meal on plasma EAA bioavailability for protein anabolism. We hypothesized that consuming a meal with two oz-eq portions of animal-based protein foods would result in greater postprandial EAA bioavailability compared to plant-based protein foods (primary) and would not elicit differential EAA bioavailability responses between young and older adults (secondary). This research will serve as an important resource for future DGAs to reevaluate the appropriateness of equating different protein sources comprising the protein foods group on an oz-eq basis for young and older adults. 

## 2. Materials and Methods

### 2.1. Ethics

The study protocols and materials were approved by the Purdue University Biomedical Institutional Review Board (IRB protocol #1804020520 (young adults) and IRB-2019-354 (older adults)), and all participants provided written, informed consent, and received monetary compensation for their participation. Before participant recruitment, these studies were registered at clinicaltrials.gov as NCT03649568 (young adults) and NCT04243395 (older adults). The reporting of this research followed the CONSORT reporting guidelines [12].

### 2.2. Participants

Participants were recruited from Greater Lafayette, IN, USA. Study inclusion criteria were: aged 22–39 or 55–75 years for young and older adults, respectively; BMI 20–35 kg/m^2^; weight stable (±4.5 kg for the previous three months); not currently (or within three months pre-study) following a moderate to vigorous intensity exercise regimen of >3 h/wk; not acutely ill; non-smoking; not diabetic; not pregnant or lactating; willing to limit purposeful physical exercise for 48 h before each testing day; willing and able to consume study foods and beverages and travel to testing facilities. Exclusion criteria for this study included: age < 22 and >39 or <55 and >75 years for young and older adults, respectively; BMI < 20 or >35 kg/m^2^; weight change > 4.5 kg within three months pre-study; exercising moderately to vigorously for >3 h/wk; being diabetic (fasting blood glucose > 126 mg/dL); being a smoker. Testing of young adults was done between February 2019 and March 2020, and testing of older adults was done between March 2020 and December 2021. 

### 2.3. Study Design

Each participant completed four 300-min trials (randomized, investigator-blinded, crossover design), with each trial separated by a minimum of three days. Participant treatment order randomization was performed by a clinical laboratory manager, who did not participate in data analysis or interpretation, using online software (Randomization.com). While the participants, clinical laboratory manager, and dietitians were not blinded, the investigators were blinded until all participants finished the protocol and all sample analyses were completed. Before each testing day, participants were asked to refrain from performing physical activity for 48 h. Participants were provided a controlled meal to consume to satiation the evening before each testing day (1052 kcal: fat: 25.5 g; carbohydrate: 163.6 g; protein: 47.2 g) (Appendix A), with no other food intake before testing the next day. On each of the four testing days, participants returned to the laboratory testing facility after a minimum 10-hour overnight period of fasting, and a catheter was placed into an antecubital vein. Baseline fasting blood samples were drawn immediately (0 min) before consumption of a standardized meal and two oz-eq of (1) unprocessed lean pork loin, (2) scrambled whole eggs, (3) black beans, or (4) raw sliced almonds. Additional postprandial blood samples were drawn at 30, 60, 120, 180, 240, and 300 min from the start of the meal. The experimental design protocol is shown in Figure 1.

### 2.4. Test Meals

On each of the four testing days, participants consumed a carefully portioned test meal (Appendix A) and two oz-eq of (1) unprocessed lean pork loin, (2) scrambled whole eggs, (3) black beans, or (4) raw sliced almonds. The energy and macronutrient contents for the test meal and two oz-eq of each Protein Food are provided in Table 1. The pork used in the study was purchased from a local butcher (Purdue Butcher Block, West Lafayette, IN, USA) and came from the same animal. The scrambled whole eggs were prepared uniformly from large eggs based on the American Egg Board’s recommendation (https://www.aeb.org/foodservice/egg-safety-handling/preparation-guidelines, accessed on 1 November 2018). The canned black beans and almonds were all purchased with the same batch numbers (US Foods, 9332313 and 3604147 for the black beans and almonds, respectively). The black beans were drained, rinsed with water, and the surface water allowed to dry before being weighed into a two oz-eq portion. All trial meals were developed by a registered dietitian using Pronutra software version 3.3 (Viocare, Inc., Princeton, NJ, USA). All foods were procured, prepared, portioned, and provided to the participants by research staff in the Department of Nutrition Science Metabolic Kitchen at Purdue University, West Lafayette, IN, USA.

### 2.5. Sample Collection and Biochemical Analyses

Blood samples were collected into serum- and plasma-separator tubes at the specified time points, centrifuged for 15 min at 4000 rpm and 4 °C, aliquoted into 1 mL microcentrifuge tubes, and stored at −80 °C as previously described [13]. Serum samples were analyzed for glucose by photometric assay (COBAS Integra 400 Analyzer; Roche Diagnostic Systems, Indianapolis, IN, USA). Insulin was assayed using an electroluminescence immunoassay on a COBAS e411 analyzer (Roche Diagnostic Systems, Indianapolis, IN, USA). Plasma amino acid (AA) concentrations were determined using High-Performance Liquid Chromatography (Agilent 1260 Infinity High-Performance Autosampler, Agilent Technologies, Santa Clara, CA, USA) with the Agilent Amino Acid Protocol [14]. Twenty AAs were reported (Ala, Arg, Apn, Asp, Cys, Gln, Glu, Gly, His, Ile, Leu, Lys, Met, Phe, Pro, Ser, Thr, Trp, Tyr, and Val).

### 2.6. Statistical Analysis

Previous research from our group with a similar design and with EAA positive incremental area under the curve (iAUCpos) as the primary outcome showed a between-trial, within-subject variability of ±2000 μg/mL/h [13]. Applying this variability to the power calculation for the current studies indicated that *n* = 30 adults would provide a sufficient sample size to detect statistical significance (α ≤ 0.05) and avoid type II errors (β ≥ 80%) for a differential EAA iAUCpos of 350 ± 2000 μg/mL/h (effect size f = 0.175). Statistical analyses of TAA, EAA, BCAA, leucine, glucose, and insulin were done for (1) young adults and (2) older adults using a doubly repeated-measures analysis of variance to compare four experimental trials and seven sampling time points. Group least-squares means (±SE) at each time point were compared using the difference of least-squares means. iAUCpos were calculated using the trapezoidal rule [15]. A repeated-measures analysis of variance was used to compare the iAUCpos among trials. Statistical significance was assigned with a Bonferroni-adjusted *p* < 0.05. As there were no differential EAA iAUCpos responses between young and older adults (secondary hypothesis), the data from these two cohorts were pooled to provide greater power for the primary hypothesis to assess differences among the four protein foods (primary hypothesis). Exploratory analyses included investigating the effect of sex on baseline (fasting) and postprandial AAs, glucose, and insulin responses. All statistical analyses were completed using SAS 9.4 (SAS Institute Inc., Cary, NC, USA).

## 3. Results

### 3.1. Randomization and Participant Characteristics

Details of the study enrollment and conduct for this project are described in Figure 2.

#### 3.1.1. Young Adults

Of the 39 potential participants who completed an initial in-person screening, 37 were enrolled and randomized, and 30 (15 females and 15 males) completed the study. Seven participants withdrew from the study: two due to starting a new job and relocating; two due to personal reasons; two chose to discontinue after the first trial; and one due to challenges placing intravenous catheter lines. The mean (±SD) age of participants was 26.0 (±4.9) years, BMI was 26.4 (±4.5) kg/m^2^, and 14 were White non-Hispanic/Latino, nine were Hispanic/Latino, three were Black/African American, three were Asian, and one was Native Hawaiian or Other Pacific Islander.

#### 3.1.2. Older Adults

Of the 30 potential participants who completed an initial in-person screening, 27 were enrolled and randomized, and 25 (15 females and 10 males) completed the study. Two participants chose to discontinue after the first trial because of the time commitment. The mean (±SD) age of participants was 64.2 (±6.6) years, BMI was 26.1 (±3.7) kg/m^2^, 24 were White non-Hispanic/Latino, and 1 was Asian.

For this project, there were no adverse events reportable to the Purdue University Institutional Review Board.

### 3.2. Baseline Fasting Blood Concentrations

Among the four trials, baseline fasting TAA, EAA, BCAA, leucine, glucose, and insulin were not different for young adults, older adults, or combined. Among the four trials, baseline fasting EAA, BCAA, and leucine were greater for young than older adults, with no age-related differences in TAA, glucose, or insulin.

### 3.3. Essential Amino Acids

#### 3.3.1. Young Adults

A main effect of protein oz-eq food sources was observed for EAA iAUCpos. Essential amino acid iAUCpos were greater for pork or eggs than for black beans or almonds; greater for pork than eggs; and not different between black beans and almonds (Appendix A and Figure 3C). Time point analysis identified greater EAA for pork or eggs than for black beans at 30, 60, 120, 180, 240, and 300 min; greater for pork or eggs than for almonds at 60, 120, 180, and 240 min; peak EAA for both pork and eggs at 120 min; greater for pork than for eggs at 120 min; and greater for pork than almonds at 300 min (Appendix A and Figure 3A).

#### 3.3.2. Older Adults

A main effect of protein oz-eq food sources was observed for EAA iAUCpos. Essential amino acid iAUCpos was greater for pork or eggs than for black beans or almonds; greater for pork than for eggs; and not different between black beans and almonds (Appendix A and Figure 3C). Time point analysis identified greater EAA for pork or eggs than for black beans or almonds at 60, 120, 180, and 240 min; peak EAA for both pork and eggs at 120 min, with pork being greater than eggs at 120 min; and greater for pork than for almonds at 300 min (Appendix A and Figure 3B).

#### 3.3.3. Young vs. Older Adults

Essential amino acid iAUCpos was not different between young and older adults overall or among trials (Figure 3C). Time point analysis identified greater EAA for young than older adults: for pork at 30, 60, and 120 min; for eggs at 30, 60, 120, and 180 min; for black beans at 30, 60, 120, 180, and 240 min; and for almonds at 30, 60, 120, 180, 240, and 300 min.

#### 3.3.4. Young and Older Adults Combined

A main effect of protein oz-eq food sources was observed for EAA iAUCpos. Essential amino acid iAUCpos was greater for pork or eggs than black beans or almonds, greater for pork than eggs, and not different between black beans and almonds (Appendix A and Figure 4B). Time point analysis identified greater EAA for pork or eggs than for black beans at 30 min; greater EAA for pork or eggs than for black beans or almonds at 60, 120, 180, 240, and 300 min; and peak EAA for both pork and eggs at 120 min, with pork being greater than eggs at 120 and 180 min (Appendix A and Figure 4A).

### 3.4. TAA, BCAA, and Leucine

The responses and statistical results (i.e., comparisons among trials and each time point) for TAA, BCAA, and leucine were comparable to EAA for young adults, older adults, young vs. older adults, and combined (Appendix A and Appendix A).

### 3.5. Serum Glucose and Insulin

#### 3.5.1. Young Adults

No differences were observed for glucose or insulin iAUCpos overall or among trials (Appendix A and Appendix A). Glucose and insulin were greater for black beans than eggs at 60 min (Appendix A and Appendix A).

#### 3.5.2. Older Adults

No differences were observed for glucose iAUCpos overall or among trials (Appendix A and Appendix A) or for glucose at any time point among trials (Appendix A and Appendix A). No differences among trials were observed for insulin iAUCpos, although it trended toward significance (*p* = 0.055).

Insulin iAUC trended toward being greater for black beans than almonds (*p* = 0.068). (Appendix A and Appendix A). At 60 min, insulin was greater for pork or black beans than for almonds and lower for eggs than for black beans (Appendix A and Appendix A).

#### 3.5.3. Young vs. Older Adults

A main effect of age was observed for glucose iAUCpos. Glucose iAUCpos was lower for young than older adults for black beans but not for pork, eggs, or almonds (Appendix A). No time-point differences for glucose were identified among trials between young and older adults.

Insulin iAUCpos was not different between young and older adults overall or among trials (Appendix A). Time point analysis identified that insulin was greater for young than older adults for eggs, black beans, or almonds at 30 min and for pork at 60 min.

#### 3.5.4. Young and Older Adults Combined

There was no main effect of protein oz-eq food source on glucose iAUCpos, although it trended toward significance (*p* = 0.082). Glucose iAUCpos was not different among trials, although glucose iAUCpos trended toward being greater for pork than for almonds (*p* = 0.083) (Appendix A and Appendix A). Time point analysis identified that glucose was greater for black beans than for pork or eggs at 60 min and trended toward being greater than almonds at 60 min (*p* = 0.064) (Appendix A and Appendix A).

A main effect of protein oz-eq food sources was observed for insulin. Insulin iAUCpos was not different among trials, although it trended toward being greater for black beans than for almonds (*p* = 0.059) (Appendix A and Appendix A). Time point analysis identified that insulin was greater for pork or black beans than for eggs or almonds at 60 min (Appendix A and Appendix A).

### 3.6. Females vs. Males

The TAA, EAA, BCAA, leucine, glucose, and insulin concentrations for each time point and iAUCpos are reported in Appendix A for females and males, respectively.

#### 3.6.1. Baseline Fasting Blood Concentrations

Baseline fasting EAA, BCAA, and leucine were lower for females than males, with no sex-specific differences in TAA, glucose, or insulin.

#### 3.6.2. Plasma Amino Acids

A main effect of sex and treatment × sex was observed for EAA iAUCpos. Essential amino acid iAUCpos were greater for females than males overall and for pork, but not for eggs, black beans, or almonds (Figure 5C). Time point analysis identified that EAA were lower for females than for males for pork, eggs, black beans, or almonds at 30 and 60 min, and black beans at 120 min. The responses and statistical results for TAA, BCAA, and leucine were comparable to those for EAA (Appendix A and Appendix A).

#### 3.6.3. Serum Glucose and Insulin

Glucose and insulin iAUCpos were not different between females and males overall or among trials (Appendix A). No time-point differences for glucose were identified among trials between sexes. Time-point analysis identified that insulin was lower for females than males for almonds at 60 min. No other time point differences for insulin were identified between sexes among trials.

## 4. Discussion

To the best of the author’s knowledge, this is the first project to assess the effect of consuming the same oz-eq portions of protein foods, namely pork, eggs, black beans, and almonds, as defined by the DGA, in the context of a mixed meal on EAA substrate bioavailability for protein anabolism in young and older adults. Consistent with our hypotheses, consuming a meal with two oz-eq of unprocessed lean pork or whole eggs resulted in greater postprandial EAA bioavailability compared to a meal with two oz-eq of black beans or raw sliced almonds in (1) young adults; (2) older adults; and (3) young and older adults combined. No differences were observed between young and older adults. In addition, lean pork resulted in greater EAA bioavailability than eggs in young adults and older adults, separately and combined. These findings show that, on the same oz-eq basis, consuming these animal- vs. plant-based protein foods more effectively provides bioavailable EAA for protein anabolism.

These findings are consistent with research by Park et al. [6], who investigated the anabolic responses to consuming two oz-eq portions of varied protein foods (not as part of a meal) in young adults. The authors found animal-based protein foods—namely pork loin, whole eggs, and beef sirloin—resulted in greater postprandial EAA and whole-body net protein balance (anabolic response) compared to plant-based protein foods—namely tofu, kidney beans, peanut butter, and mixed nuts. The magnitude of the anabolic responses was positively associated with the EAA content of each food and the postprandial EAA responses. The EAA iAUCpos responses to consuming the same two oz-eq portions of pork, eggs, beans, and nuts shown in our current project and by Park et al. [6] are comparable. Based on these observations, future research is warranted to assess whether the association between EAA iAUCpos and whole-body net protein balance shown in young adults would also occur in older adults.

Taken together, the findings from our project and others [5,6] indicate the protein quality of a food or meal (i.e., the EAA content of a meal) is a primary determinant of postprandial EAA bioavailability and, subsequently, the stimulus for muscle and whole-body protein anabolism [5,6,7]. These results are pertinent to the DGA’s call for a shift to more plant-based eating patterns [1]. While there are health benefits to consuming more plant-based foods [8,10], guidance on the importance of protein/AAs nutrition—of which animal-based protein-rich foods are high-quality protein/AAs sources—for muscle and whole-body health across the lifespan to promote healthy aging is important information to incorporate when providing dietary recommendations [16].

Consistent with our secondary hypothesis, we did not observe differential EAA iAUCpos responses between young and older adults overall or among protein foods. There is consistent evidence that basal (fasting) protein synthesis does not differ between young and older adults [17,18,19,20,21]. Therefore, the impaired retention of lean tissue masses in older adults may partially result from an attenuated anabolic response to ingesting dietary protein/AAs [17,18,22,23]. Differences in protein/AA digestion, absorption, and tissue uptake may alter the concentrations of circulating AAs. Age-related differences in these processes may have contrasting and potentially off-setting effects on circulating AA concentrations. For example, greater splanchnic tissue AAs uptake [24,25] and slower gastric emptying [26] in older adults may promote lower postprandial AAs concentrations relative to young adults. In contrast, smaller plasma volumes [27] due to lower total body water and a reduced rate of peripheral tissue uptake of AAs from circulation [28] in older adults could promote higher postprandial AAs concentrations relative to younger adults. It is likely a combination of these factors contributed to why we did not see differential EAA iAUCpos responses between young and older adults.

Our observation that postprandial EAA iAUCpos were greater for females than for males for pork may be due to the females consuming more protein relative to body mass and plasma volume. However, limited evidence suggests the anabolic response to the same protein/AA ingestion or infusion is not different between young [29] and middle-aged [30] adult females and males. In contrast, older females have been shown to exhibit a lower postprandial anabolic response to ingestion of the same protein as older males [31].

Regarding project strengths and limitations, this is the first project to assess the effects of consuming the same oz-eq portions of protein foods as defined by the DGA on postprandial EAA responses in the context of mixed whole food meals in both young and older adults. We chose to test a two oz-eq portion of each Protein Food to theoretically obtain measurable postprandial EAA responses. We are mindful that this specific portion likely does not reflect the amounts of these protein foods consumed on a meal-to-meal or weekly basis by young and older adults. Importantly, with consideration of protein quantity, EAA content and bioavailability, and disparate metabolizable energy contents, these results highlight the shortcomings of using oz-eq to achieve the DGA recommendations for protein foods. Our combined sample size of 55 (30 females, 25 males) participants exceeded our a priori estimated need based on effect size, and we used conservative Bonferroni-adjusted *p* values to assign statistical significance. While not preferred, recruitment and testing of the young and older participants were conducted consecutively, albeit with the same randomized crossover experimental design and procedures. Future comparisons of age effects with young and older participants should be integrated into the same randomization and study design. Research supports greater postprandial EAA responses to protein feeding being associated with greater muscle and whole-body anabolism [7]. However, our research did not include direct measures of acute or chronic changes in muscle protein synthesis or whole-body protein balance.

## 5. Conclusions

In conclusion, based on the oz-eq concept used in the DGAs, the animal- and plant-based protein foods included in this project do not equivalently provide bioavailable EAA for protein anabolism in young and older adults. Whole eggs and especially unprocessed lean pork consumed with a standard meal resulted in greater postprandial EAA bioavailability than black beans or raw sliced almonds. This research demonstrates that the word “equivalent” in the oz-eq unit of measure for protein foods does not apply to the protein content and postprandial essential amino acid bioavailability of these foods.

## Figures and Tables

**Figure 1 nutrients-15-02870-f001:**
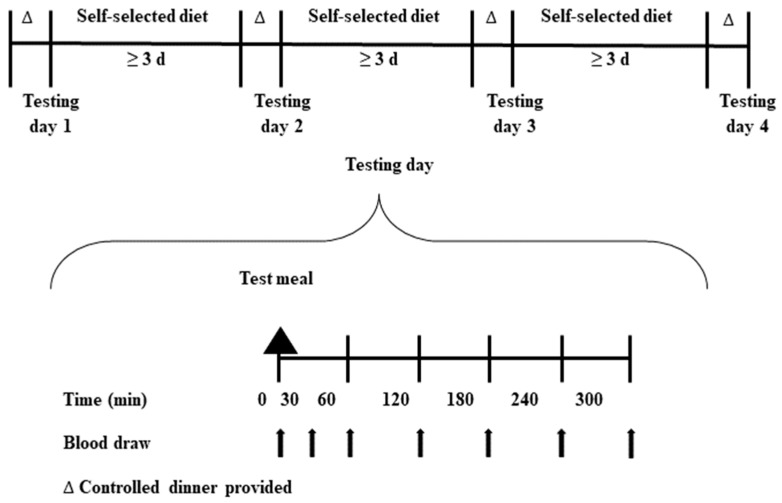
Experimental design protocol.

**Figure 2 nutrients-15-02870-f002:**
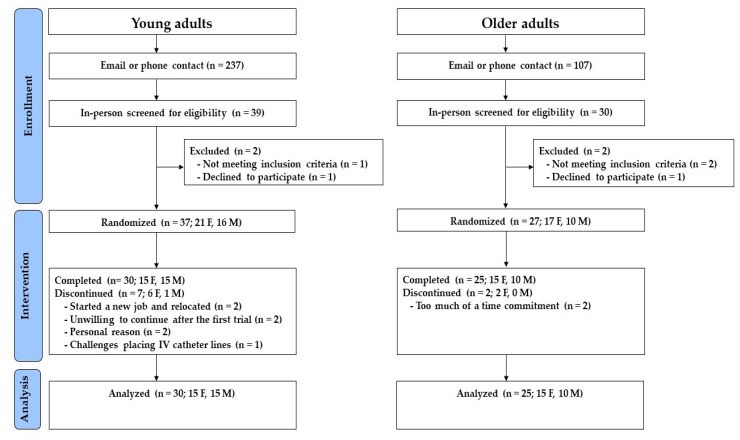
Participant recruitment flow diagram for young and older adults.

**Figure 3 nutrients-15-02870-f003:**
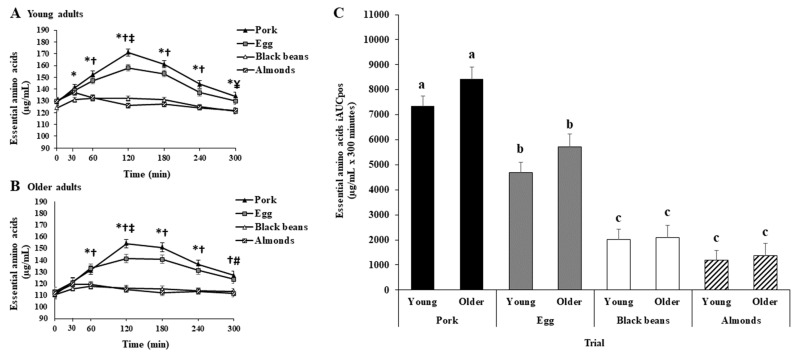
Plasma essential amino acid concentrations for young adults (**A**) and older adults (**B**) in the fasted (0-min) and postprandial (300-min) periods after initiation of meal consumption for pork, eggs, black beans, and almonds. Different symbols indicate a significant difference between or among trials at each time point (Bonferroni adjusted, *p* < 0.05). * denotes a significant difference between pork or eggs vs. black beans. † denotes a significant difference between pork or eggs vs. almonds. ‡ denotes a significant difference between pork and eggs. ¥ denotes a significant difference between pork and almonds. # denotes a significant difference between pork and black beans. (C) Essential amino acids with positive incremental area under the curve (iAUCpos) for pork (black bars), eggs (gray bars), black beans (white bars), and almonds (dashed black and white bars) for young and older adults. Different letters indicate a significant difference among trials within an age group (*p* < 0.05). All values are least squares means ± SE; young adults: *n* = 30 (15 females, 15 males); older adults: *n* = 25 (15 females, 10 males).

**Figure 4 nutrients-15-02870-f004:**
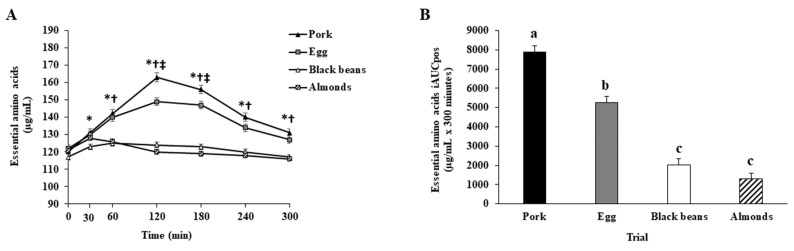
(**A**) Plasma essential amino acid concentrations for young and older adults combined in the fasted (0-min) and postprandial (300-min) periods after initiation of meal consumption for pork, eggs, black beans, and almonds. Different symbols indicate a significant difference between or among trials at each time point (Bonferroni adjusted, *p* < 0.05). * denotes a significant difference between pork or eggs and black beans. † denotes a significant difference between pork or eggs and almonds. ‡ denotes a significant difference between pork vs. eggs. (**B**) Essential amino acids: positive incremental area under the curve (iAUCpos) for pork (black bars), eggs (gray bars), black beans (white bars), and almonds (dashed black and white bars) for young and older adults combined. Different letters indicate a significant difference among trials (*p* < 0.05). All values are least squares means ± SE; *n* = 55 (30 (15 females, 15 males) young adults and 25 (15 females, 10 males) older adults).

**Figure 5 nutrients-15-02870-f005:**
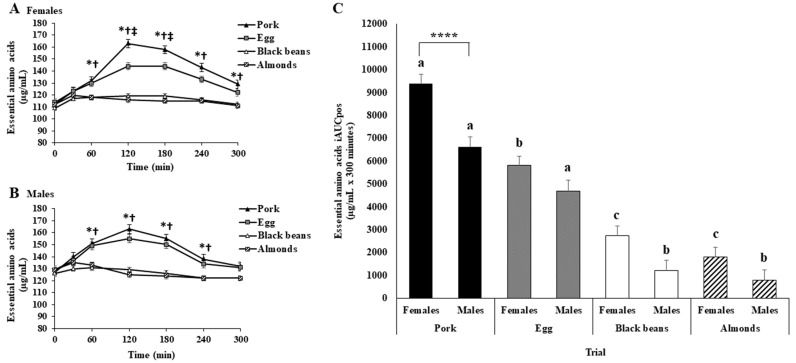
Plasma essential amino acid concentrations for females (**A**) and males (**B**) in the fasted (0-min) and postprandial (300-min) periods after initiation of meal consumption for pork, eggs, black beans, and almonds. Different symbols indicate a significant difference between or among trials at each time point (Bonferroni adjusted, *p* < 0.05). * denotes a significant difference between pork or eggs vs. black beans. † denotes a significant difference between pork or eggs and almonds. ‡ denotes a significant difference between pork and eggs. (**C**). Plasma essential amino acids have a positive incremental area under the curve (iAUCpos) for pork (black bars), eggs (gray bars), black beans (white bars), and almonds (dashed black and white bars) for females and males. Different letters indicate a significant difference among trials within an age group (Bonferroni adjusted *p* < 0.05). **** denotes a significant difference between females and males for pork (Bonferroni adjusted *p* < 0.0001). All values are least squares means ± SE; females: *n* = 30; males: *n* = 25.

**Table 1 nutrients-15-02870-t001:** Energy and macronutrient contents of the test meal and protein food sources, and essential amino acid content of the protein food sources ^1^.

	Energy (kcal)	Fat (g)	CHO (g)	Protein (g)	EAA (g)
Test meal	218	11.5	25.8	6	2.09
Lean pork loin (2 oz-eq)	73	1	0	14	7.36
Whole eggs (2 oz-eq)	145	10	0	12.5	5.38
Black beans (2 oz-eq)	113	0.5	20	7.5	3.02
Almonds (2 oz-eq)	161	14	6	6	1.85

^1^ The quantity of protein for each trial includes the protein from the test meal: lean pork (20 g); whole eggs (18.5 g); black beans (13.5 g); and almonds (12 g). The total quantity of EAA for each trial includes the EAA from the test meal: lean pork loin (9.45 g); whole eggs (7.47 g); black beans (5.11 g); and almonds (3.94 g). EAA—essential amino acids.

## Data Availability

The data supporting this project’s findings are available from the corresponding author, W.W.C., upon reasonable request.

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
