# Peer review of "Effects of Consuming Ounce-Equivalent Portions of Animal- vs. Plant-Based Protein Foods, as Defined by the Dietary Guidelines for Americans on Essential Amino Acids Bioavailability in Young and Older Adults: Two Cross-Over Randomized Controlled Trials"

_nutrients, 2023, doi:10.3390/nu15132870_

Round 1

Reviewer 1 Report

In the manuscript submitted to me for review entitled: Effects of Consuming Ounce-Equivalent Portions of Animal- Versus Plant-Based Protein Foods, as Defined by the Dietary Guidelines for Americans on Essential Amino Acids Bioavailability in Young and Older Adults: Cross-Over Randomized Controlled Trials the authors: Gavin Connolly , Joshua L. Hudson , Robert E. Bergia , Eric M. Davis , Austin S. Hartmann , Wenbin Zhu , Chad C. Carroll , Wayne W. Campbell assessed the effects of consuming two ounce-equivalent (oz-eq) servings of pork, eggs, black beans, and almonds and the available essential amino acids (EAA) content after a meal by tracking bioavailability in young and older adults.

In the introduction, the authors present the main food sources of proteins and their impact on the body after consumption.

In the Materials and Methods section, the participants included in the study and the study design itself are well described. To support their study, the authors used information from 31 references, which covered data summarized for more than 35 years (in the period from 1986 to 2022). Of the cited references, 8 are from the last 5 years (about 1/4 of the total number). No redundant self-quotes are used.

With today's increasing obesity in society, such research is of great importance and can contribute to the proper preparation of new protein diets, nutrition control and a healthier lifestyle. 

My remarks and recommendations to the authors are: 

1.     Should Figure 2 be bold on line 180? 

2.     I would recommend if possible to increase the size of the text in Figure 2. It is a bit difficult to read when presented like this.

3.     The following figures are presented in the Supplementary file: S3, S4, S5, Fig 6, Fig 7, Fig 8, Fig 9, Fig 10, S11, S12, S13, S14, S15 and S16. They present the main information of the results obtained by the authors. They are very well presented and I think they should be placed in the main text of the manuscript. In this version, it is difficult to perceive the information they carry.

4. General note regarding the same pass in different places:

on line 113 it is noted to see Table S1;

on line 124 - Table S2;

on line 213 - Table S3;

on line 222 - Table S4;

on line 248 - Table S5;

on line 300 - Tables S6-S7.

However, these tables are not presented either in the manuscript or in the Supplementary file. Let them be supplemented, and according to my recommendation, their place is also in the main text of the manuscript. 

5.In the Supplementary Materials section at the end of the manuscript, Supplementary Figures and Supplementary Tables (which are generally missing) are described. In my opinion, as I indicated in my above remarks, they should be placed in the manuscript itself, because they present the results obtained by the authors, and separating them in Supplementary Materials makes it difficult to read and perceive the information from the manuscript. 

6. In the section References in literary sources with numbers 19 and 21, not all authors are represented. Let all the authors be added.

7. Reference #11 does not have an exact year of publication listed. Let it be pointed out.

Reviewer 2 Report

Here the authors report the results of two cross-over randomized controlled trials that compared the effects of consuming two ounce-equivalent portions of animal-based (pork or eggs) versus plant-based (black beans or almonds) protein foods on essential amino acids bioavailability in young and older adults. The main finding was that animal-based protein foods resulted in greater postprandial essential amino acids bioavailability than plant-based protein foods in both age groups, and that pork had greater bioavailability than eggs. The authors suggest that the same ounce-equivalent portions of protein foods as defined by the Dietary Guidelines for Americans do not provide equivalent essential amino acids content and bioavailability for protein anabolism in young and older adults. The paper is well-written, and fits within the scope of the journal. It should also interest a broad readership. The figures are clear and the statiscal tests accurate.

Minor point:

The study did not include direct measures of muscle or whole-body protein synthesis or balance, which are important indicators of protein anabolism. This could be discussed in the discussion.

The study used a single dose of two ounce-equivalent portions of protein foods, which may not reflect the typical dietary patterns or protein intake of young and older adults. This could also be discussed in the discussion part. 
